# Reliability, validity, and responsiveness of the Thai version of the Dry Eye-Related Quality-of-Life Score questionnaire

**Napaporn Tananuvat**[1]*, **Sasiwimon Tansanguan**[1], **Nahathai Wongpakaran**[2], **Tinakon Wongpakaran**[2]

1 Faculty of Medicine, Department of Ophthalmology, Chiang Mai University, Chiang Mai, Thailand,
2 Faculty of Medicine, Department of Psychiatry, Chiang Mai University, Chiang Mai, Thailand

* ntananuvat@gmail.com

## Abstract

Dry eye disease (DED) is a common and growing eye problem worldwide. Chronic DED symptoms can, subsequently, affect the patients' quality of life (QOL). This prospective cross-sectional study aimed to assess the reliability, validity, and responsiveness of the Thai version of the Dry Eye-Related Quality-of-Life Score (DEQS-Th) questionnaire and to evaluate its accuracy in DED screening. Psychometric validation was conducted on DED participants. All participants completed the DEQS-Th and other measurements including the Ocular Surface Disease Index (OSDI) and the 5-level EQ-5D (EQ-5D-5L). Internal consistency, concurrent validity, convergent, and discriminant validity were evaluated. The standardized response mean (SRM) was used to evaluate the responsiveness of the DEQS-Th. The optimal cut-off score of DEQS-Th for DED screening was assessed. Among 100 participants with a mean age of 50.9 ± 14.4 years, and 89.0% female, the internal consistency of the DEQS-Th was excellent (Cronbach's alpha: 0.80–0.92). The test-retest intra-class correlation was 0.82–0.92. It showed concurrent validity with the OSDI (r = 0.694, p < .001) and EQ-5D-5L index scores (r = -0.578, p < .001). DED is suspected if the DEQS-Th score ≥ 18.33 (AUC = 0.897, sensitivity 90.0%, specificity 76.7%) or its Short Form score ≥ 3 (AUC 0.857, sensitivity 93.0%, specificity 63.3%). The SRM of the symptom subscale of DEQS-Th was 0.82, indicating relatively large responsiveness, whereas the impact on daily life subscale and the summary score was small. In conclusion, the DEQS-Th is valid and reliable for evaluating the multifaceted effects of DED on a patient's QOL. It can be useful for primary assessment and monitoring of DED in routine clinical practice.

## Introduction

Dry eye disease (DED) is characterized by a vicious cycle of tear film instability and hyperosmolarity leads to ocular surface inflammation and damage, and neurosensory abnormalities [1]. It is a common eye problem worldwide and one of the frequent causes of patients seeking eye care practitioners [2]. DED prevalence is expected to be increased further because of the

**Data Availability Statement:** All relevant data are within the paper and its Supporting Information file.

**Funding:** Yes. This research was funded by the Faculty of Medicine, Chiang Mai University Endowment Fund, grant number 010-2562 to N.T. The funders had no role in the design of the study; in the collection, analyses, or interpretation of data; in the writing of the manuscript, or in the decision to publish the results.

**Competing interests:** The authors have declared that no competing interests exist.

increased use of visual display terminals, increased aging population, and stressful socio-environment [3]. Chronic DED symptoms of ocular discomfort and visual impairment are associated with limitations in performing daily activities, decreased quality of life (QOL), work productivity, and economic burden [4–8].

According to the International Dry Eye Workshop 2017(DEWS II) report, the diagnosis of DED involves the presence of symptoms assessed by a dry eye questionnaire as well as any of three specific signs including tear break-up time (TBUT), tear osmolarity, and ocular surface staining [9]. Likewise, the DED diagnosis proposed by the Asia Dry Eye Society in 2016 was based on subjective symptoms and reduced TBUT [10]. In this regard, the quantification of subjective symptoms is one of the primary methods of DED diagnosis.

Even though symptoms and signs of dry eye are inconsistent and vary across individuals and DED sub-types [11,12], the ability to accurately quantify ocular symptoms is important for screening patients who may need additional evaluation. It is also critical to monitor the disease progression and treatment response. However, it is difficult to standardize and quantify patients' symptoms in clinical settings. Thus, it is important to accurately assess ocular symptoms associated with DED, its impact on everyday function, or health-related QOL through valid and reliable questionnaires.

Currently, several dry eye questionnaires have been used in gathering symptoms in clinical research and practice [9]. However, six dry eye questionnaires have questions on health-related QOL and have been evaluated for psychometric properties including the Ocular Surface Disease Index (OSDI), the Impact of Dry Eye on Everyday Living (IDEEL), the 25-item National Eye Institute's Visual Function Questionnaire (NEI VFQ-25), the Dry Eye–Related Quality-of-Life Score (DEQS), the University of North Carolina Dry Eye Management Scale (UNC DEMS) and the Chinese version of the Dry Eye–Related Quality of Life (CDER QOL) [13]. All of these questionnaires have different purposes and limitations. The OSDI has 12 questions, which renders it comparatively short with a low time burden. However, it is difficult to determine whether it allows comprehensive all dry eye symptoms and health-related QOL considerations. The IDEEL has 57 questions with comprehensive coverage that takes a long time to complete and is not free for use. The NEI VFQ-25 is a questionnaire that can be applied to various ophthalmic diseases and conditions. However, its reliability and validity in DED remain unclear. The DEQS which was developed in Japan in 2013, has good reliability, validity, and responsiveness in evaluating the effects of DED on daily living [14], while the UNC DEMS [15] and CDER QOL [16] are relatively new questionnaires developed according to the U.S. Food and Drug Administration guidance [17]. The three latest questionnaires have not been validated in other languages.

Despite the high prevalence of DED in Thailand [18], limited measurements exist to assess both symptoms and health-related QOL in Thai. Our preliminary study on the Thai version of the DEQS (DEQS-Th) demonstrated that this tool is valid and user-friendly in the assessment of dry eye symptoms among normal study samples [19]. This present study aimed to determine the psychometric properties of the DEQS-Th questionnaire including its reliability, validity, and responsiveness in DED patients. Also, the sensitivity and specificity of the DEQS-Th for DED screening were conducted.

## Materials and methods

This prospective cross-sectional study was approved by the Institute Review Board before being initiated (study code: OPT-2561-005562) and followed the Declaration of Helsinki. All volunteers signed a written informed consent after a complete explanation.

## Study participants

One hundred participants, diagnosed as DED at the Ophthalmology Clinic, Chiang Mai University Hospital, were recruited between 2018–2019. The eligible criteria included adult subjects aged ≥ 18, voluntary participation, and literate in Thai. The criteria for diagnosis of DED complied with those defined by the DEWS II. Table 1 shows the inclusion and exclusion criteria for participant selection.

All participants underwent complete ophthalmic examination for both eyes including measuring of visual acuity, intraocular pressure, and additional dry eye tests such as corneal fluorescein staining (CFS), TBUT, and basic tear secretion (Schirmer's test). CFS scores were assigned based on a modified van Bijsterveld grading system (the average score was the mean of the sum scores of the 3-area nasal, mid, and temporal cornea ranging from 0 [none] to 3 [maximum]) [20]. TBUT was measured using fluorescein staining without anesthesia. The participant was asked to blink several times. The interval between the last complete blink and the first dry spot on the cornea was measured and the average of three consecutive TBUT was recorded. Schirmer's test was performed with anesthesia. After drying the excess tears, the Schirmer strip was placed at the lateral one-third of the lower fornix for five minutes. The strip was then removed and the wetting length of the filter paper was measured in mm.

## Procedure

All DED participants were asked to complete the DEQS-Th questionnaire and additional health-related QOL questionnaires including the OSDI and the 5-level EuroQol-5-Dimensions (EQ-5D-5L).

To evaluate the reproducibility, the DED participants completed the DEQS-TH twice. The re-test was performed two weeks after the first test.

To evaluate the responsiveness or change of the questionnaire regarding the response to treatment, ten patients received diquafosol tetrasodium 3% ophthalmic solution six times/day for treatment of DED. This eye drop is a purinergic P2Y2 receptor agonist on the ocular surface. It stimulates both water and mucin secretion from conjunctival epithelial cells and goblet cells, thereby rehydrating the ocular surface independent of tear secretion from the lacrimal glands [21–23]. Participants were asked to complete the DEQS-Th before and at four weeks (± 3 days) after diquafosol treatment.

**Table 1. Participant selection criteria.**

| Inclusion criteria | Exclusion criteria |
|---|---|
| • Age ≥ 18 years<br>• Criteria for DED diagnosis [a]<br> 1. Ocular symptoms (OSDI score ≥ 13)<br> 2. Tear film abnormality [b]<br>  (1) TBUT ≤ 5 seconds<br>  (2) Schirmer test with anesthesia < 5 mm<br> 3. Ocular surface abnormality: corneal fluorescein staining [c]<br>• Literate in the Thai Language | • BCVA worsen than 6/18<br>• Ocular infection or inflammation<br>• Ocular surgery within 6 months<br>• Systemic diseases or disabilities that affect daily life activities |

BCVA, best-corrected visual acuity; DED, dry eye disease.

[a] The criteria was modified according to those defined by the International Dry Eye Workshop 2017 by Tear Film and Ocular Surface Society [9].

[b] The tear function was considered abnormal if (1) or (2) applied.

[c] Abnormality of the ocular surface was designated as a positive corneal fluorescein staining (scores range 1–3).

## Instruments

**Dry Eye-related Quality-of-Life Score.** The DEQS questionnaire contains 15 questions divided into two subscales: Bothersome Ocular Symptoms (6 questions) and Impact on Daily Life (9 questions). Each question has columns A and B for the frequency and severity, respectively. Response to the frequency portion in column A is based on a 5-point scale ranging from "none of the time" (0) to "all of the time" (4). A frequency score of 1–4 points prompts the respondent to proceed to the severity in column B to answer regarding the degree of severity on a four-point scale. The DEQS score is calculated with the following formula: (sum of the degree scores for all questions answered) x 25/ (total number of questions answered). The score "0" indicates the best possible score (no symptoms) and "100" indicates the worst possible score (maximum symptoms) [14].

**Thai version of the Dry Eye-related Quality-of-Life Score.** The DEQS-Th was developed from the English version of the DEQS questionnaire [14]. In brief, after permission from the owner of the DEQS (the Asia Dry Eye Society and Santen Pharmaceutical Co., Japan.), the translation and cross-cultural adaptation of the DEQS questionnaire into Thai was conducted according to principles of good practice reported by the International Society for Pharmacoeconomics and Outcomes Research (ISPOR) [24]. Language translation and cultural adaptation, then preliminary psychometric validation with 30 normal participants were performed in our previous study [19].

**Ocular Surface Disease Index.** The OSDI consists of 12 questions with 3 subscales: ocular symptoms, vision-related function, and environmental triggers. Each DED patient rated symptoms on a 5-point scale from 0 (never) to 4 (always). The OSDI total score was obtained by multiplying the sum scores of all questions answered by 25 and dividing by the total number of questions answered, giving the OSDI a scale from 0–100 with higher scores reflecting greater disability. According to the OSDI scores, the patients were classified as normal (scores 0–12), mild (13–22), moderate (23–32), and severe (33–100) symptoms [25]. The Thai version of OSDI was used in this study in compliance with the English version of the OSDI (Allergan Inc., Irvine, CA, USA).

**EuroQol-5-Dimensions 5-Level.** The EQ-5D, developed by EuroQoL, is composed of five items concerning 'mobility´, 'self-care, 'usual activities´, 'pain/discomfort´, and 'anxiety/depression [26]. It is a 5-point Likert scale, ranging from 0 (no problem) to 5 (unable/extreme problems). The EQ-5D has two parts, 1) a descriptive system that calculates a five-digit code specifying a specific health state to the index score. The score ranges from 0 to 1, with 0 meaning death and 1 meaning complete health. However, the index score can also have a negative value, meaning worse than dead, 2) and a visual analog scale (EQ-VAS), ranging from 0 (worst imaginable health state) to 100 (best imaginable health state). The Thai version EQ-5D-5L and the index score were used in this study [27].

## Statistical analysis

The participants' demographic data were descriptively analyzed. The CFS, TBUT, and Schirmer test from the worse eye was used for data analysis. For numerical data, the mean (SD) was used for data with normal distribution, while the median (range) was used for non-normally distributed data. The internal consistency was calculated to evaluate the reliability of the questionnaire; Cronbach's alpha coefficient $\geq 0.7$ was considered acceptable. Intraclass correlation coefficient (ICC) was calculated to determine the temporal relationship in test-retest reliability. Concurrent validity was evaluated using Pearson's correlation coefficient to evaluate the correlations between the DEQS-Th scores and other measurements including the OSDI and EQ-5D-5L index scores.

Convergent validity is denoted by the level of correlation between constructs and instruments. These relations may be strong or weak correlations depending on the relationship expected between the constructs or instruments compared [28]. We created a correlation matrix between the QOL assessed by the subscale Impact on Daily Life scores of the DEQS-Th and scores from EQ-5D. Correlations would be expected to be high if the similar domain of impact of daily life and EQ-5D were assessed, thereby demonstrating convergent validity.

Discriminative validity was conducted to assess whether a measure can discriminate between the groups [28]. The total scores of DEQS-Th were analyzed to compare normal and clinical samples to indicate its discriminatory ability.

Responsiveness is defined as an ability of a measurement to detect clinically significant changes over time [29]. It was evaluated by comparing the DEQS-Th scores at baseline and follow-up periods after treatment using the standardized response mean (SRM). SRM values of 0.8, 0.5, and 0.2 were considered to be large, moderate, and small, respectively [30].

Floor or ceiling effects indicate the limitation of content validity, and reliability of the questionnaire. Floor or ceiling effects were suggested to be no more than 15% [31]; otherwise, it may affect responsiveness as the participants' changes cannot be assessed [29].

To find the optimal cut-off score of DEQS-Th for suspected DED, the gold standard diagnosis for dry eye was made by using the OSDI score of $\geq$ 13 and the TBUT of $\leq$ 5 seconds. The receiver operating characteristic (ROC) curve was generated and the area under the ROC (AUC) was analyzed to determine the accuracy of the DEQS-Th. Sensitivity, specificity, positive predictive value, negative predictive value, and estimated cost were calculated. To simply apply in real-life practice, we also evaluated the Short Form DEQS-Th (SF DEQS-Th) for DED screening by using a sum of frequency scores of subscale Bothersome Ocular Symptoms. Our previous published data from non-DED participants were served as a control in some parts of the analysis [19]. A p-value $< 0.05$ was used to determine the significant level. SPSS program (version 22.0, SPSS Inc., Chicago, IL, USA) was used for data analysis.

## Results

### Participants

Among100 DED participants, 89% were females with a mean age of 50.9 ± 14.4 (20–84) years. The participants' demographic data were demonstrated in Table 2. According to the OSDI grading severity, the dry eye symptoms were classified as a mild-to-moderate degree in 30% and a severe degree in 70% of the patients. The mean OSDI score was 46.8 ± 21.4 for all DED participants. No floor or ceiling effects were found in the total scores of the DEQS-Th.

### Psychometric analysis

**Item analysis.**   The mean frequency and degree score of each item, as well as the total DEQS-Th score, were significantly higher in DED participants than in non-DED participants indicating discriminant validity (Tables 2 and 3).

**Reliability.**   The results of internal consistency were demonstrated in Table 4. Cronbach's alpha for the frequency and degree score of the subscale ocular symptoms, impact on daily life, and summary score range from 0.81 to 0.92. The two-week test-retest reliability was evaluated in 90 DED participants. The ICC ranged from 0.80 to 0.92, indicating excellent reproducibility.

**Concurrent and convergent validity.**   Concurrent Validity: The DEQS-Th had a significantly positive correlation with OSDI (r = 0.694, p < 0.001) and each subscale score of the OSDI (r = 0.449 to 0.672, p < 0.001). The DEQS-Th showed a significantly negative correlation with EQ-5D-5L index scores (r = -0.578, p < 0.001), indicating concurrent validity

**Table 2. Demography and characteristics of DED and non-DED participants.**

| Characteristics | DED (n = 100) | Control** (n = 30) | p-value |
|---|---|---|---|
| Age: mean (SD) | 50.9 (14.4) | 38.6(12.9) | < .001[a] |
| range, years | 20–84 | 22–60 | |
| Gender: Female, n (%) | 89 (89.0%) | 23 (76.7%) | .086[a] |
| Ocular diseases, n (%) | 80 (80.0%) | 0 (0%) | < .001[a] |
| • DED | 75(75%) | | |
| • Cataract | 10(10%) | | |
| • Pterygium | 10(10%) | | |
| • Pinguecula | 2(2%) | | |
| • Glaucoma | 1(1%) | | |
| Systemic diseases, n (%) | 65 (65.0%) | 10 (33.3%) | .008[a] |
| • Hypertension | 23(23%) | 2(6.7%) | |
| • Dyslipidemia | 18(18%) | 3(10%) | |
| • Allergy | 16(16%) | - | |
| • Systemic lupus erythematosus | 8(8%) | 2(10%) | |
| • Diabetic mellitus | 5(5%) | 10(33%) | |
| • Inactive cancer | 5(5%) | - | |
| • Hypothyroidism | 4(4%) | - | |
| • Osteoporosis | 2(2%) | 1(3.3%) | |
| • Miscellaneous* | 8(8%) | - | |
| Regular exercise, n (%) | 42 (42.0%) | 5 (16.7%) | .011[b] |
| Wearing contact lens, n (%) | 4 (3.8%) | 0 (0%) | .266[a] |
| Smoking, n (%) | 2 (2.0%) | 2 (6.7%) | .228[a] |
| DEQS-Th score: mean (SD) | | | |
| Ocular symptoms subscale | 18.5 (6.1) | 9.3(7.9) | < .001[b] |
| Impact on daily life subscale | 45.4 (22.0) | 15.4(15.7) | < .001[b] |
| Summary score | 43.7 (19.8) | 14.8(12.7) | < .001[b] |

DED, dry eye disease; DEQS-Th, the Thai version of the Dry Eye-Related Quality-of-Life Score.

* Gout (2), anemia (1), chronic kidney disease (1), coronary artery disease (1), gastroesophageal reflux disease (1), migraine headache (1), polycystic ovarian syndrome (1).

**Data from our previous published study in normal participants [19].

[a] The χ2 test was used for statistical comparisons

[b] The Mann-Whitney U test was used for statistical comparisons.

(Table 5). However, the DEQS-Th scores were not correlated with any clinical dry eye tests including TBUT, CFS, and Schirmer's test.

Convergent Validity: Table 6 shows the correlation matrix between the QOL assessed by the subscale Impact on Daily Life of the DEQS-Th and scores from EQ-5D items, depression item of DEQS-Th significantly related to anxiety/depression items of EQ-5D, higher than other dimensions. The items of eye functions such as opening eyes, blurred vision, and sensitivity to bright light were related more to mobility, self-care, and usual activity of the EQ-5D. The items of QOL that were related to function such as reading, watching, and studying were, as expected, significantly related to the physical function domain of EQ-5D rather than the anxiety/depression domain. Items "Feeling distracted" and "Not feeling like going out" were related to both physical and anxiety/depression as they were involved in concentration. All significant correlations indicate convergent validity.

**Clinical validity and responsiveness.** The responsiveness was evaluated in ten patients receiving diquafosol treatment. At four weeks after treatment, the symptom scores of the DEQS-Th and CFS were significantly improved and the SRM were 0.816 and 1.061, respectively (Table 7).

**Table 3. Mean score of each item of the DEQS-Th between DED and non-DED participants.**

| Group Statistics | Frequency | | p-value[a] | Degree | | p-value[a] |
|---|---|---|---|---|---|---|
| | DED | Control* | | DED | Control* | |
| *Bothersome Ocular Symptoms* | | | | | | |
| 1. Foreign body sensation | 2.12(0.97) | 1.03(1.00) | < .001 | 2.21(0.87) | 1.03 (1.03) | < .001 |
| 2.Dry sensation in eyes | 2.52(0.96) | 0.80(0.93) | < .001 | 2.49(0.86) | 0.73(0.83) | < .001 |
| 3. Painful or sore eyes | 1.00(1.12) | 0.27(0.58) | .001 | 1.15(1.25) | 0.30(0.65) | < .001 |
| 4.Ocular fatigue | 1.77(1.15) | 0.80(0.93) | < .001 | 1.85(1.15) | 0.73(0.83) | < .001 |
| 5. Heavy sensation in eyelids | 1.17(1.19) | 0.40(0.81) | .001 | 1.20(1.21) | 0.30(0.60) | < .001 |
| 6.Redness in eyes | 0.84(1.00) | 0.20(0.41) | .001 | 0.96(1.16) | 0.23(0.50) | .001 |
| *Impact on Daily Life* | | | | | | |
| 1. Difficulty in opening eyes | 1.00(1.15) | 0.10(0.40) | < .001 | 1.18(1.31) | 0.10(0.40) | < .001 |
| 2. Blurred vision when watching something | 2.40(0.99) | 1.17(1.09) | < .001 | 2.50(0.98) | 1.33(1.16) | < .001 |
| 3. Sensitivity to bright light | 2.21(1.12) | 0.80(0.96) | < .001 | 2.37(1.17) | 0.87(1.04) | < .001 |
| 4. Problems with eyes when reading | 2.20(1.13) | 0.83(1.09) | < .001 | 2.29(1.07) | 0.97(1.19) | < .001 |
| 5. Problems with eyes when watching television or looking at a computer or cell phone | 2.28(1.06) | 0.77(0.86) | < .001 | 2.38(1.05) | 0.87(0.97) | < .001 |
| 6. Feeling distracted because of eye symptoms | 1.64(1.19) | 0.47(0.68) | < .001 | 1.78(1.25) | 0.50(0.73) | < .001 |
| 7. Eye symptoms affecting work | 1.81(1.22) | 0.60(0.97) | < .001 | 1.94(1.25) | 0.67(1.03) | < .001 |
| 8. Not feeling like going out because of eye symptoms | 1.15(1.21) | 1.70(0.46) | < .001 | 1.28(1.29) | 0.23(0.63) | < .001 |
| 9. Feeling depressed because of eye symptoms | 0.55(0.91) | 0.10(0.40) | .010 | 0.64(1.03) | 0.10(0.40) | .006 |

DED, dry eye disease; DEQS-Th, the Thai version of the Dry Eye-Related Quality-of-Life Score.

[a] The Mann-Whitney U test was used for statistical comparisons

*Adapted from a previously published study in normal participants [19].

## Accuracy for DED screening

In predicting DED against the gold standard, we used the AUC as the criterion to compare the following set of items. The scale provided AUCs of 0.897 (p<0.001, 95%CI = 0.831 to 0.943) denoting good accuracy performance [32] (Fig 1A). The DEQS-Th yielded a sensitivity of 90.00 and specificity of 76.67% for the cut-off score ≥ 18.33, based on Youden's index [33] (S1 Table). The AUCs of the SF DEQS-Th was 0.857 (p<0.001, 95% CI 0.785–0.912) and the optimal cut-off value was ≥ 3, which yielded a sensitivity of 93.0% and specificity of 63.3% (Fig 1B, S2 Table).

## Discussion

This study demonstrates the reliability and validity of the DEQS-Th questionnaire in dry eye patients. Our preliminary study found that the DEQS-Th has good internal consistency when

**Table 4. Internal consistency and reproducibility of the DEQS-Th.**

| Subscale | Internal consistency: Cronbach's alpha (N = 100) | | Test-retest: Reproducibility (ICC) (N = 90) | |
|---|---|---|---|---|
| | Frequency | Degree | Frequency | Degree |
| Ocular symptoms | 0.81 | 0.82 | 0.80 | 0.82 |
| Impact on daily life | 0.92 | 0.91 | 0.91 | 0.90 |
| Summary score | 0.92 | 0.92 | 0.92 | 0.92 |

DEQS-Th, the Thai version of the Dry Eye-Related Quality-of-Life Score; ICC, Intraclass Correlation Coefficient.

**Table 5. Correlation between DEQS-Th, OSDI, index score of EQ-5D, and clinical tests among DED patients.**

| Correlations | Eye Symptoms | | Impact on Daily life | | Summary Scores | |
|---|---|---|---|---|---|---|
| | r | p | r | p | r | p |
| *OSDI* | 0.588** | < .001 | 0.686** | < .001 | 0.694** | < .001 |
| • Ocular symptoms | 0.606** | < .001 | 0.641** | < .001 | 0.672** | < .001 |
| • Vision-related function | 0.471** | < .001 | 0.604** | < .001 | 0.593** | < .001 |
| • Environmental triggers | 0.449** | < .001 | 0.556** | < .001 | 0.551** | < .001 |
| *EQ-5D-5L index scores* | -0.535** | *< .001* | -0.543** | *< .001* | -0.578** | *< .001* |
| *Clinical dry eye tests* | | | | | | |
| • TBUT | -0.046 | .646 | -0.050 | .625 | -0.052 | .609 |
| • Schirmer's test | 0.076 | .600 | 0.147 | .307 | 0.131 | .363 |
| • CFS | 0.046 | .648 | 0.023 | .824 | 0.034 | .740 |

CFS, Corneal fluorescein staining; DED, dry eye disease; DEQS-Th, the Thai version of the Dry Eye-Related Quality-of-Life Score; EQ-5D-5L, EuroQol-5-Dimensions 5-Level; OSDI, Ocular Surface Disease Index; TBUT, tear break-up time.

tested in normal participants (Cronbach alpha values of the Bothersome Ocular Symptoms, the Impact on Daily Life, and the summary scores were 0.71, 0.88, and 0.89, respectively) [19]. When tested in DED patients, the DEQS-Th has excellent internal consistency and test-retest reliability similar to those of the original version of DEQS (Cronbach's alpha = 0.83 to 0.93 and ICC = 0.81 to 0.93) [14].

In addition, the DEQS-Th questionnaire has been shown to correlate well with the OSDI, which is a current gold standard. The OSDI is one of the most frequently used tools in DED assessment and has good psychometric properties in the assessment of subjective dry eye symptoms and their effects on visual-related activities within the previous week [25]. This study demonstrated that the DEQS-Th questionnaire has a criterion (concurrent) validity with the OSDI, consistent with a related study by Inomata et al. that demonstrated a strong correlation between the DEQS and the Japanese version of OSDI (J-OSDI) scores. They found that the J-OSDI scores tended to be higher than the DEQS (31.6 vs. 27.6) which was in accordance with our study [34].

**Table 6. Correlation matrix between the quality of life assessed by the subscale Impact of Daily Life scores and scores from EQ-5D-5L items among DED patients.**

| Subscale Impact on Daily Life | Mobility | Self-care | Usual activities | Pain/discomfort | Anxiety/depression |
|---|---|---|---|---|---|
| Difficulty in opening eyes | 0.521** | 0.462** | 0.381** | 0.339** | 0.413** |
| 2. Blurred vision when watching something | 0.195 | 0.332** | 0.382** | 0.249* | 0.219* |
| 3. Sensitivity to bright light | 0.202* | 0.313** | 0.298** | 0.087 | 0.206* |
| 4. Problems with eyes when reading | 0.322** | 0.390** | 0.266** | 0.191 | 0.203* |
| 5 Problems with eyes when watching television or looking at a computer or cell phone | 0.150 | 0.245* | 0.343** | 0.210* | 0.200* |
| 6. Feeling distracted because of eye symptoms | 0.250* | 0.356** | 0.371** | 0.249* | 0.352** |
| 7. Eye symptoms affecting work | 0.326** | 0.444** | 0.454** | 0.248* | 0.319** |
| 8. Not feeling like going out because of eye symptoms | 0.315** | 0.419** | 0.377** | 0.397** | 0.408** |
| 9. Feeling depressed because of eye symptoms | 0.335** | 0.487** | 0.469** | 0.368** | 0.559** |

DED, dry eye disease; EQ-5D-5L, EuroQol-5-Dimensions 5-Level.

*Correlation is significant at the 0.05 level

** Correlation is significant at the 0.01 level.

**Table 7. Mean and standardized response mean of the DEQS-Th scores evaluated in patients treated with diquafosol (n = 10).**

| Variables | Mean (SD) | | p-value | SRM |
|---|---|---|---|---|
| | Before treatment | After treatment | | |
| *DEQS-Th scores* | | | | |
| Bothersome Ocular Symptoms | 43.75 (21.72) | 29.17 (14.03) | .025[a] | 0.816 |
| Impact on Daily Life | 45.00 (29.01) | 38.33 (22.79) | .333[b] | 0.258 |
| Summary score | 44.50 (24.98) | 34.67 (18.39) | .169[b] | 0.453 |
| *Clinical dry eye tests* | | | | |
| TBUT (sec) | 3.50 (1.27) | 4.30 (2.06) | .196[a] | 0.480 |
| Schirmer's test (mm) | 5.30 (4.06) | 5.55 (3.48) | .759[b] | 0.066 |
| CFS (score 0–3) | 1.00 (0.82) | 0.31 (0.48) | .024[b] | 1.061 |

DEQS-Th, the Thai version of the Dry Eye-Related Quality-of-Life Score; TBUT, tear break-up time; CFS, corneal fluorescein staining; SRM, standardized response mean.

[a] An Independent Samples t-test was used for statistical comparisons

[b] The Wilcoxon Signed Ranks test was used for statistical comparisons.

Chronic DED can have multi-faceted effects on a patient's health including personality [35] and psychosomatic symptoms [7,36]. A previous study found that OSDI scores significantly correlated with the DEQS, anxiety, depression, and stress scores. Moreover, the severity of DED symptoms impacted more on the depressive symptoms [36]. Recently, a large-scale study using the mobile application suggests that depressive symptoms are more common in individuals with more severe dry eye symptoms [7]. The original DEQS questionnaire has been reported to correlate well with the mental components of the NEI VFQ-25 [14]. In this study, the QOL assessed by the subscale Impact on Daily Life of the DEQS-Th demonstrated convergent validity with the EQ-5D-5L. It is noted that all impact on QOL items is significantly related to the self-care and usual activities of the EQ-5D-5L, while most items are related to anxiety/depression indicating that DED influenced both the function and mental health of each affected individual. Nevertheless, DEQS-Th scores did not correlate with clinical signs of dry eye. This finding agrees with previous studies regarding the discordance between symptoms and signs of DED [11,12].

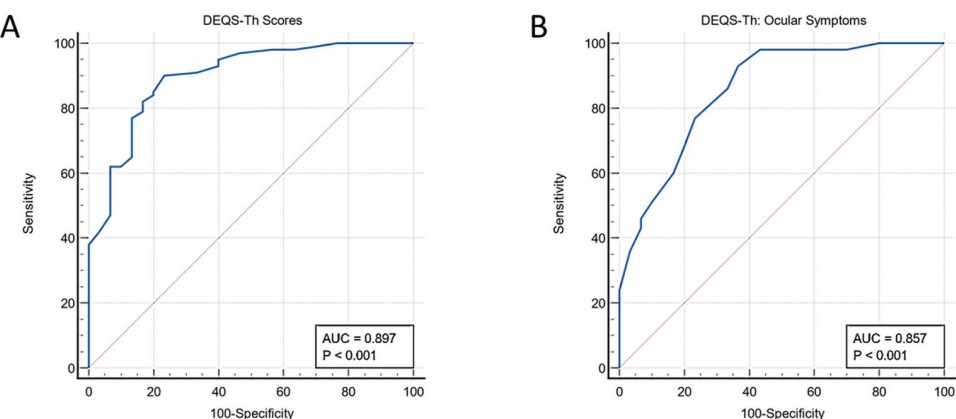

**Fig 1.** Receiver operating characteristic curve of the DEQS-Th scores (A) and the Short Form DEQS-Th scores (B) for the prediction of dry eye disease diagnosis. DEQS-TH; the Thai version of the Dry Eye-Related Quality-of-Life Score.

The discriminant validity of the DEQS-Th was verified from the finding that all DEQS-Th scores were significantly higher in the DED than those of the control groups. Our findings are in accordance with Sakane's study even though the average scores of subjects with DED and non-DED (33.7 vs. 6.0) were lower than those in our study [14]. The reasons may be due to the difference in the study population. Besides, both convergent and discriminant validity are evidence of the construct validity of the DEQS-Th questionnaire.

This study also examined the effect of treatment by using the DEQS-Th. We found that the symptom scores and CSF significantly improved after four weeks of diquafosol treatment. Although other parameters improved after receiving diquafosol, they did not reach statistical significance. Our results comply with a previous study using the DEQS questionnaire to evaluate the effects of diquafosol in DED [37]. They found that diquafosol improved both symptoms and signs in DED patients. Compared to Sakane's study, the DEQS scores and clinical signs (TBUT and fluorescein staining) significantly improved after treatment with a punctal plug [14]. The different results may be because the effects of topical medication like diquafosol may take more time than performing a lacrimal punctal occlusion. With a greater number of the study samples, diquafosol might have significantly changed the DEQS-Th scores and TBUT. Nevertheless, our findings suggest that DEQS-Th was useful in assessing the changes in DED symptoms and the therapeutic effect.

In predicting performance, DEQS-Th is shown to have good accuracy for detecting DED compared to the OSDI (AUC = 0.897 and 0.744 for DEQS and OSDI, respectively) [38]. This study also demonstrated the cut-off score of the DEQS-Th of $\geq 18.33$ for screening DED. Practically, a cut-off score of 18 can be applied even though the values of sensitivity and specificity might be slightly changed. This cut-off value is higher than the value of $> 15$ which was previously proposed by Ishikawa [39]. Ishikawa's study was conducted among 333 soldiers (mostly males), while most of the subjects in our study were relatively older females with more severe dry eye symptoms. The different cut-off values might be due to the variation of the patient demographics such as sex, age, and ethnicity. In addition, this study also provided the cut-off score of the SF DEQS-Th by using the subscale ocular symptoms for DED screening and the value of $\geq 3$ was the optimal criterion. The SF DEQS-Th is simple and can, further, be widely used in screening individuals with suspected DED. Thus, the burden for the physicians in real-life practice can be lowered. However, caution should be applied to diagnose DED based on symptoms only because the discrepancy between symptoms and signs from asymptomatic patients often occurs [12]. Therefore, it is important to use both the questionnaire and clinical examination for a holistic assessment of DED.

The strength of this study is a comprehensive assessment of the DEQS-Th, providing robust evidence in terms of its psychometric properties and performance prediction for detecting DED. Some limitations need to be mentioned. First, the study was conducted in a university hospital, tertiary care setting with a relatively high prevalence, compared to that of the general population with most patients experiencing a more severe degree of DED. Second, other factors such as medication, depression, anxiety, and environmental effects were not accounted for. This may have affected the results. Last, the sample size for testing responsiveness was small.

## Conclusions

The DEQS-Th is valid and reliable for evaluating both dry eye symptoms and their impact on a patient's QOL. It demonstrates psychometric properties that can be useful for primary evaluation and monitoring of DED in clinical practice and research. This study also provides the cut-off score of the DEQS-Th for screening individuals with suspected DED, thus preventing morbidity from untreated disease.

## Supporting information

**S1 Table. Criterion values of the DEQS-Th and coordinates of the ROC curve.** DEQS-Th, the Thai version of the Dry Eye-Related Quality-of-Life Score; ROC, the receiver operating characteristic.
(DOCX)

**S2 Table. Criterion values of the Short Form DEQS-Th and coordinates of the ROC curve.** DEQS-Th, the Thai version of the Dry Eye-Related Quality-of-Life Score; ROC, the receiver operating characteristic.
(DOCX)

**S1 Dataset. Dataset of this study.**
(XLSX)

## Acknowledgments

The authors would like to thank Mrs. Supaporn Martin for her kind providing language help.

## Author Contributions

**Conceptualization:** Napaporn Tananuvat, Sasiwimon Tansanguan, Nahathai Wongpakaran, Tinakon Wongpakaran.

**Data curation:** Napaporn Tananuvat, Nahathai Wongpakaran, Tinakon Wongpakaran.

**Formal analysis:** Napaporn Tananuvat, Sasiwimon Tansanguan, Nahathai Wongpakaran, Tinakon Wongpakaran.

**Funding acquisition:** Napaporn Tananuvat.

**Methodology:** Napaporn Tananuvat, Sasiwimon Tansanguan, Nahathai Wongpakaran.

**Project administration:** Napaporn Tananuvat.

**Validation:** Tinakon Wongpakaran.

**Writing – original draft:** Napaporn Tananuvat, Sasiwimon Tansanguan.

**Writing – review & editing:** Napaporn Tananuvat, Nahathai Wongpakaran, Tinakon Wongpakaran.

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
