## [Decision Letter · Decision Letter 0]

18 May 2022

PONE-D-22-00892Reliability, validity, and responsiveness of the Thai version of the Dry Eye-Related Quality-of-Life Score questionnairePLOS ONE

Dear Dr. Tananuvat,

Thank you for submitting your manuscript to PLOS ONE. After careful consideration, we feel that it has merit but does not fully meet PLOS ONE’s publication criteria as it currently stands. Therefore, we invite you to submit a revised version of the manuscript that addresses the points raised during the review process.

We look forward to receiving your revised manuscript.

Kind regards,

Adrienne Csutak, MD, PhD, MSc

Academic Editor

PLOS ONE

Journal Requirements:

2. We noted in your submission details that a portion of your manuscript may have been presented or published elsewhere. Please clarify whether this publication was peer-reviewed and formally published. If this work was previously peer-reviewed and published, in the cover letter please provide the reason that this work does not constitute dual publication and should be included in the current manuscript.

Reviewers' comments:

Reviewer's Responses to Questions

**Comments to the Author**

1. Is the manuscript technically sound, and do the data support the conclusions?

Reviewer #1: Partly

2. Has the statistical analysis been performed appropriately and rigorously? 

Reviewer #1: Yes

3. Have the authors made all data underlying the findings in their manuscript fully available?

Reviewer #1: No

4. Is the manuscript presented in an intelligible fashion and written in standard English?

Reviewer #1: Yes

5. Review Comments to the Author

Reviewer #1: Review

The manuscript reports on a follow-up investigation of the author’s previous article on the Dry Eye-related Quality-of-life Score (DEQS) questionnaire translated to the Thai language, DEQS-Th, for use in Thailand. They have used statistical methods for producing quantitative measures of the effectiveness of the questionnaire for screening patients with suspected dry eye disease (DED). They compared DEQS-Th with other DED tests and evaluated reliability, internal consistency, test-retest correlation, responsiveness, and accuracy of the DEQS-Th.

The authors report cut-off scores to be used for screening and detecting DED using the DEQS-Th. Inconsistencies between the Abstract, Results and Discussion are noted below. However, there is a fundamental issue relative to the cut-off scores that should be addressed. The DEQS-Th cut-off scores are reported as “>X” for diagnostic tests and “>Y” for ocular symptoms. As can be seen from the Supplemental Tables S1 and S2, as the score rises, the sensitivity of the test decreases while the specificity increases. This is a standard property of Receiver Operator Characteristic (ROC) curves (Fig. 1). Is it preferable to have a higher cut-off with decreased sensitivity (fewer cases of DED detected) and increased specificity (fewer false positives)? Or is it better to have a lower cut-off with increased sensitivity (more cases of DED detected) and decreased specificity (more false positives)? The authors should acknowledge that using a cut-off value represents a trade-off between sensitivity and specificity. If they recommend using “>” with the cut-off, they should justify the choice being made between sensitivity and specificity.

Some specific comments:

Lines 30-31, 281-282, 344: The cut-off scores, “>18.33“ and “>19”, for rating the DEQ-Th diagnostic tests are not equivalent. As can be seen in the Supplementary S1 Table, there is a difference between a cut-off of 18.33 and any other cut-off value. Only 18.33 gives the sensitivity of 90 and specificity of 76.67. For any other cut-off number, the sensitivity and specificity will be different. To report a cut-off of 19, the authors should recalculate the sensitivity and specificity corresponding to the cut-off and report those results in lines 281-282. In any event, the reported cut-off, sensitivity and specificity in the Abstract (lines 30-31) and the cut-off in the Discussion (line 344) need to be changed to be consistent with the results reported in the Results on lines 281-282. In addition, “>” is inappropriate if the reported cut-off is supposed to represent a particular sensitivity and specificity.

Lines 31-32, 283-284, 350: The cut-off scores, “>3” and “>4”, for rating the DEQ-Th using subscale ocular symptoms for DED screening are not equal. As can be seen in the Supplementary S2 Table, there is a difference between a cut-off of 3 and a cut-off of 4. A cut-off of 3 gives the sensitivity equal to 93.00 and specificity equal to 63.33, while a cut-off of 4 gives a sensitivity of 86.00 and a specificity of 66.67. The cut-off, sensitivity and specificity given in the Abstract (lines 31-32) and the cut-off in the Discussion (line 350) need to be consistent with the cut-off, sensitivity and specificity given in the Results in lines 283-284. As noted above, here too, “>” is inappropriate if the reported cut-off is supposed to represent a particular sensitivity and specificity.

Lines 73: The DEQS was developed in 2013, not 2014.

Lines 148-154: For the OSDI, the patients use a 5-point scale, 0-4. Need to specify how this is translated to a 0-100 scale – to obtain the total OSDI score, calculate the sum of scores of all questions answered times 25 divided by total number of questions answered, giving the OSDI a scale from 0-100.

Lines 209, 220, 226: none-DED should be non-DED

Lines 421-423: Add the date of the FDA document, December 2009, to Ref 17 and update the availability date from 2021 to be current in 2022.

Supplementary Material: The results presented in this manuscript, Tables 2-7 and S1 and S2, are based on data collected in the course of this research. The data have not been made available with this submission.

6. PLOS authors have the option to publish the peer review history of their article (what does this mean?). If published, this will include your full peer review and any attached files.

Reviewer #1: No

---

## [Author Response · Author response to Decision Letter 0]

24 May 2022

Response to Reviewers

Dear Reviewer,

We greatly appreciate your time for your kind review and constructive comments on the manuscript entitled “Reliability, validity, and responsiveness of the Thai version of the Dry Eye-Related Quality-of-Life Score questionnaire” (Submission ID PONE-D-22-00892). 

We have carefully studied your comments and made corrections as suggested. 

Please find enclosed the revised version of our manuscript highlighted in yellow and the detailed point-by-point response to the comments raised by the reviewer.

We hope that our responses would be satisfactory.

Sincerely yours,

Napaporn Tananuvat, M.D.

(On behalf of the co-authors)

Reviewer #1: Review

The manuscript reports on a follow-up investigation of the author’s previous article on the Dry Eye-related Quality-of-life Score (DEQS) questionnaire translated to the Thai language, DEQS-Th, for use in Thailand. They have used statistical methods for producing quantitative measures of the effectiveness of the questionnaire for screening patients with suspected dry eye disease (DED). They compared DEQS-Th with other DED tests and evaluated reliability, internal consistency, test-retest correlation, responsiveness, and accuracy of the DEQS-Th.

The authors report cut-off scores to be used for screening and detecting DED using the DEQS-Th. Inconsistencies between the Abstract, Results and Discussion are noted below. However, there is a fundamental issue relative to the cut-off scores that should be addressed. The DEQS-Th cut-off scores are reported as “>X” for diagnostic tests and “>Y” for ocular symptoms. As can be seen from the Supplemental Tables S1 and S2, as the score rises, the sensitivity of the test decreases while the specificity increases. This is a standard property of Receiver Operator Characteristic (ROC) curves (Fig. 1). Is it preferable to have a higher cut-off with decreased sensitivity (fewer cases of DED detected) and increased specificity (fewer false positives)? Or is it better to have a lower cut-off with increased sensitivity (more cases of DED detected) and decreased specificity (more false positives)? The authors should acknowledge that using a cut-off value represents a trade-off between sensitivity and specificity. If they recommend using “>” with the cut-off, they should justify the choice being made between sensitivity and specificity.

Response: Thank you for your comments and suggestions. We apologize for the inconsistencies. The data are now corrected. We strongly agree with you that there is a trade-off between sensitivity and specificity using the different cut-off scores. That’s why we use a cut-off of ≥18.33 that yields a sensitivity of 90.00% and a specificity of 76.67%. This optimal criterion also provides the lowest cost (0.131) when the cost estimation is taken into account. In the revised manuscript, we apply “≥” for the criterion values in both Supplement S1 Table and S2 Table as suggested.

Some specific comments:

Lines 30-31, 281-282, 344: The cut-off scores, “>18.33“ and “>19”, for rating the DEQ-Th diagnostic tests are not equivalent. As can be seen in the Supplementary S1 Table, there is a difference between a cut-off of 18.33 and any other cut-off value. Only 18.33 gives the sensitivity of 90 and specificity of 76.67. For any other cut-off number, the sensitivity and specificity will be different. To report a cut-off of 19, the authors should recalculate the sensitivity and specificity corresponding to the cut-off and report those results in lines 281-282. In any event, the reported cut-off, sensitivity and specificity in the Abstract (lines 30-31) and the cut-off in the Discussion (line 344) need to be changed to be consistent with the results reported in the Results on lines 281-282. In addition, “>” is inappropriate if the reported cut-off is supposed to represent a particular sensitivity and specificity.

Response: Thank you for your careful review. In the revised manuscript, the cut-off score of ≥ 18.33 was used consistently in the Abstract, the Results, and the Discussion part. 

Practically, we suggest a cut-off score of ≥ 18 for simple use of the DEQS-TH questionnaire even though the exact sensitivity and specificity may be slightly changed from the original 18.33.

This statement is added in the Discussion part (lines 346-347).

Lines 31-32, 283-284, 350: The cut-off scores, “>3” and “>4”, for rating the DEQ-Th using subscale ocular symptoms for DED screening are not equal. As can be seen in the Supplementary S2 Table, there is a difference between a cut-off of 3 and a cut-off of 4. A cut-off of 3 gives the sensitivity equal to 93.00 and specificity equal to 63.33, while a cut-off of 4 gives a sensitivity of 86.00 and a specificity of 66.67. The cut-off, sensitivity and specificity given in the Abstract (lines 31-32) and the cut-off in the Discussion (line 350) need to be consistent with the cut-off, sensitivity and specificity given in the Results in lines 283-284. As noted above, here too, “>” is inappropriate if the reported cut-off is supposed to represent a particular sensitivity and specificity.

Response: Thanks again for your comment. The cut-off score ≥ 3 was used consistently in the revised manuscript. 

Lines 73: The DEQS was developed in 2013, not 2014.

Response: Thank you for the correction. The year of the DEQS development is changed to be… “2013” (line 73).

Lines 148-154: For the OSDI, the patients use a 5-point scale, 0-4. Need to specify how this is translated to a 0-100 scale – to obtain the total OSDI score, calculate the sum of scores of all questions answered times 25 divided by total number of questions answered, giving the OSDI a scale from 0-100.

Response: The calculation of the OSDI total score is added up as suggested as… “The OSDI total score was obtained by multiplying the sum scores of all questions answered by 25 and dividing by the total number of questions answered, giving the OSDI a scale from 0-100 with higher scores reflecting greater disability” (Lines 150-152).

Lines 209, 220, 226: none-DED should be non-DED

Response: Thanks for your careful review. These typing errors are corrected. We also re-check thorough the manuscript for any other mistakes.

Lines 421-423: Add the date of the FDA document, December 2009, to Ref 17 and update the availability date from 2021 to be current in 2022.

Response: Thank you for the suggestion. This reference is edited as suggested (line 425).

Supplementary Material: The results presented in this manuscript, Tables 2-7 and S1 and S2, are based on data collected in the course of this research. The data have not been made available with this submission.

Response: We believe that all relevant data are within the paper. However, the data are enclosed with this revision.

---

## [Editor Report · Decision Letter 1]

27 Jun 2022

Reliability, validity, and responsiveness of the Thai version of the Dry Eye-Related Quality-of-Life Score questionnaire

PONE-D-22-00892R1

Dear Dr. Tananuvat,

We’re pleased to inform you that your manuscript has been judged scientifically suitable for publication and will be formally accepted for publication once it meets all outstanding technical requirements.

Kind regards,

Adrienne Csutak, MD, PhD, MSc

Academic Editor

PLOS ONE
---

## [Editor Report · Acceptance letter]

11 Jul 2022

PONE-D-22-00892R1 

Reliability, validity, and responsiveness of the Thai version of the Dry Eye-Related Quality-of-Life Score questionnaire 

Dear Dr. Tananuvat:

I'm pleased to inform you that your manuscript has been deemed suitable for publication in PLOS ONE. Congratulations! Your manuscript is now with our production department. 

Kind regards, 

on behalf of

Dr. Adrienne Csutak 

Academic Editor

PLOS ONE